# Brain Disease-Modifying Effects of Radiofrequency as a Non-Contact Neuronal Stimulation Technology

**DOI:** 10.3390/ijms26052268

**Published:** 2025-03-04

**Authors:** Shulei Sun, Junsoo Bok, Yongwoo Jang, Hyemyung Seo

**Affiliations:** 1Department of Molecular and Life Sciences, Institute for Precision Therapeutics, Center for Bionano Intelligence Education and Research, Hanyang University, 55 Hanyangdaehak-ro, Ansan 15588, Republic of Korea; 2Department of Medical and Digital Engineering, College of Engineering, Hanyang University, Seoul 04736, Republic of Korea; 3Department of Pharmacology, College of Medicine, Hanyang University, Seoul 04736, Republic of Korea

**Keywords:** non-contact neuronal stimulation (NCNS), radiofrequency electromagnetic fields (RF-EMF), cognitive improvement

## Abstract

Non-invasive, non-contact, and painless methods of electrical stimulation to enhance neural function have been widely studied in recent years, particularly in the context of neurodegenerative diseases such as Alzheimer’s disease (AD) and related dementias, which cause cognitive decline and other neurological symptoms. Radiofrequency (RF), which is a rate of oscillation in the range of 3 kHz to 300 GHz (3 THz), has been suggested as one potential non-contact neuronal stimulation (NCNS) technique for improving brain function. A new type of electrical stimulation uses a radiofrequency electromagnetic field (RF-EMF). RF exposure has been shown to modulate neural stimulation and influence various brain activities in in vitro and in vivo models. Recent studies have explored the effects of RF-EMF on human physiology, particularly in areas such as brain activity, cognition, and sleep behavior. In this review, we summarize recent findings about the effects of non-contact stimulations in in vitro studies, in vivo animal models, and human clinical cases.

## 1. Introduction

Advances in non-invasive and non-contact stimulation technologies have opened new frontiers in neuroscience, particularly for understanding and potentially treating neurodegenerative diseases such as Alzheimer’s disease (AD) and related cognitive disorders. Among them, radiofrequency electromagnetic fields (RF-EMFs) have garnered significant attention as a promising tool for modulating neural activity without direct physical contact. RF-EMFs, operating at oscillation frequencies ranging from 3 kHz to 300 GHz, can penetrate human tissues and bones due to their unique wavelength and radiation properties. This ability makes them a potential candidate for non-contact neuronal stimulation (NCNS), offering an innovative approach to enhancing brain function and addressing neurodegenerative pathologies. Emerging evidence suggests that RF-EMF exposure can influence key physiological and cognitive processes, including brain activity, cognition, sleep, and even metabolic regulation. The complex interactions between RF-EMFs and biological systems have been demonstrated in various experimental models, ranging from in vitro cellular studies to in vivo animal models and human clinical trials. These studies indicate that RF-EMFs can modulate neuronal excitability, alter glucose metabolism, and affect cognitive behaviors, potentially offering therapeutic benefits for conditions like AD. However, the effects of RF-EMFs are not universally beneficial: some studies have reported potential neurotoxic outcomes, including oxidative stress, neuroinflammation, and disrupted neural development under certain exposure conditions.

In this review, we aim to provide a comprehensive synthesis of recent findings on the effects of RF-EMF stimulation across in vitro models, in vivo animal studies, and clinical research in humans. We will explore how RF-EMFs influence neurophysiological processes such as neurogenesis, neuronal excitability, and metabolism, as well as the implications for cognitive function, behavior, and sleep. Additionally, we will discuss the potential therapeutic and adverse effects of RF-EMFs, emphasizing the need for further research to elucidate the underlying mechanisms and optimize exposure parameters for clinical applications. By summarizing current knowledge, this review seeks to shed light on the promise and challenges of RF-EMF-based interventions in neuroscience and medicine.

## 2. The Effects of Non-Contact RF-EMF Stimulation on In Vitro Model Cells

The effects of non-contact RF-EMF stimulation on various aspects of neurodegenerative diseases have been reported in in vitro models (Table 1) [1,2]. Grasso et al. reported that exposure to 900 MHz RF-EMF for 20 min enhanced nestin expression and the self-renewal of olfactory ensheathing cells, which are specialized glial stem cells. This effect was linked to improved functional recovery after injury, including axonal regeneration, which suggests a potential role for RF-EMFs in the neuroregeneration process [3]. A study using exposure to 3.0 GHz RF-EMF for 60 min at low doses (<1 W/kg specific absorption rate (SAR)) showed alterations in neuronal function: the action potential amplitude decreased, and intracellular Ca^2+^ levels increased. These changes led to depolarization of the resting membrane potential, which increased neuronal excitability and synaptic transmission [4]. In another study, exposure to 918 MHz RF-EMF (0.2 W/kg SAR) for 60 min enhanced the mitochondrial membrane potential and reduced reactive oxygen species (ROS) in primary astrocytes treated with 5 µM amyloid-beta (Aβ). These data suggest that RF-EMF could have a potentially protective role against the Aβ-induced neurodegenerative environment associated with AD [5]. Perez et al. demonstrated that exposure to 64 MHz RF-EMF (0.6 W/kg SAR) for 1 h per day for 14 days significantly reduced the levels of secreted Aβ40 (46%) and Aβ42 (36%) peptides in primary human brain cells exposed to Aβ-induced pathology. That finding suggests that RF-EMF could have a potentially therapeutic role in modifying Aβ-related pathology in AD [6].

In contrast to the neuroprotective effects reported in those in vitro models, exposure to 1800 MHz RF-EMF (4 W/kg SAR) for 3 days inhibited neurite outgrowth in embryonic neural stem cell-differentiated neurons. This exposure also downregulated the expression of neurogenic genes such as *Ngn1* and *NeuroD* and increased the levels of *Hes1*, a known inhibitor of neurogenesis. These findings suggest that RF-EMF can suppress neural development and regeneration in certain conditions [7,8]. Cholinergic SN56 cells exposed to 900 MHz RF-EMF (1 W/kg SAR) showed reduced neurite outgrowth and delayed morphological maturation compared with control cells, although neurite length and branching were unaffected, indicating that RF-EMF can impact the development of neuronal processes [9]. RF-EMF exposure did not significantly affect neuronal proliferation or viability in some cases. Cholinergic SN56 cells exposed to 900 MHz RF-EMF (1 W/kg SAR for up to 144 h) showed no significant changes in cell viability, although RF-EMF exacerbated the neurotoxic effects of hydrogen peroxide exposure [10]. In rat primary cortical neurons, similar RF-EMF exposure did not alter cell viability or proliferation, even in the presence of oxidative stress agents such as Aβ or hydrogen peroxide. These data suggest that RF-EMF exposure could alter neurogenesis and neuronal outgrowth during development. Several studies indicate that RF-EMF exposure can induce cytotoxicity and oxidative stress in various neuronal cell types. SH-SY5Y cells (human neuroblastoma cells) exposed to RF-EMF (1800 MHz, 0.23 W/kg SAR) showed increased levels of monomeric α-synuclein, a potential marker of neurodegenerative diseases, as well as oxidative stress markers such as ROS [11]. Primary cortical neurons exposed to RF-EMF (1800 MHz, 2 W/kg SAR) for 24 h exhibited increased levels of 8-hydroxyguanine (8-OHdG), a marker of oxidative damage to DNA, and a reduction in mitochondrial DNA (*mtDNA*) and *mtRNA*, indicating mitochondrial dysfunction [12]. RF-EMF exposure has been shown to upregulate genes involved in apoptotic pathways. In primary cells exposed to 1900 MHz RF-EMF for 2 h, the expression of *caspase-2*, *caspase-6*, *Asc,* and *Bax* (pro-apoptotic markers) increased, suggesting that even short-term exposure can promote apoptosis in brain cells [13]. In addition, microglial activation was observed in N9 microglial cells exposed to 2.45 GHz RF-EMF (6 W/kg SAR) for 20 min, triggering significant pro-inflammatory responses. This indicates that RF-EMF exposure can stimulate microglial activation, which could contribute to neuroinflammation, possibly exacerbating AD pathology [14]. Kim et al. demonstrated that exposure to 1950 MHz RF-EMF (6 W/kg SAR) for 2 h increased ROS production, activating the JNK signaling pathway, which leads to cell damage, particularly in the context of neurotoxicity induced by glutamate [15].

Still other studies found that RF-EMF exposure had no significant effects on cell viability or neurotoxicity. Aβ-induced cytotoxicity in HT22 cells (a neuronal cell line) was not altered by exposure to 837 MHz code-division multiple access (CDMA) or 1950 MHz wideband CDMA (W-CDMA) signals [16]. In another study to check the effect of RF-EMF, RF-EMF (1950 MHz W-CDMA, 6 W/kg SAR for 2 h/day over 3 days) did not change the expression levels of APP or BACE1 in AD-associated HT22 and SH-SY5Y cells, which suggest that short-term RF-EMF exposure might not have a major effect on these specific pathological markers [17]. Exposure of SH-SY5Y cells or N9 microglial cells to RF-EMF (935 MHz, 4 W/kg SAR for 2 or 24 h) resulted in only a transient increase in the levels of the autophagy marker ATG5 and the antioxidant glutathione (GSH), suggesting no major oxidative damage [18]. In other experiments, exposing SH-SY5Y cells to RF-EMF (2.45 GHz, 6 W/kg SAR for 20 min) did not interfere with neuronal differentiation or mitochondrial dynamics [19].

Taken together, it is clear that different RF-EMF exposure has diverse effects on in vitro cell models, including the potential to modulate neurodegenerative processes in AD such as neurogenesis, neuroinflammation, and Aβ peptide production regulation. Several studies suggest potential neuroprotective effects, such as cell renewal and a reduction in Aβ levels, whereas others highlight neuroinflammatory responses and the suppression of neurite outgrowth. Some studies showed that RF-EMF exposure induced oxidative stress, mitochondrial dysfunction, and the activation of apoptotic pathways in some neuronal cell types, but other studies found that it produced no significant changes in cell viability, proliferation, or neurotoxicity. The neuroprotective and neurotoxic effects of RF-EMF are thus complex and appear to depend on factors such as cell type, exposure conditions, and the duration of exposure. Previous studies indicate the complexity of RF-EMF interactions with cellular processes and highlight both the therapeutic and toxic effects of RF-EMF exposure, particularly in the context of neurodegenerative diseases such as AD. The data suggest the need for further research to understand the mechanisms underlying the effects of RF-EMF, particularly with respect to the variability in response across different cell types and exposure conditions.

**Table 1 ijms-26-02268-t001:** The effects of RF-EMF on in vitro model cells.

Non-Contact Stimulation Exposure					
Stimulation Type	Frequency and Intensity	Exposure Period	Cell Line	Effects	References
RF-EMF	900 MHz EMF (AM 900 MHz, CM 900 MHz)	10, 15 and 20 min	Primary cells (mouse olfactory bulbs, P2)	↑ Cytoskeletal protein expression (GFAP, vimentin, nestin; CW 900 MHz for 15–20 min) ↓ Cytoskeletal protein expression (GFAP, vimentin, nestin; AM 900 MHz for 15–20 min) ↑ Caspase-3 expression (AM 900 MHz for 20 min)	[3]
	3.0 GHz 0.3/0.7 W/kg SAR	60 min	Primary hippocampal neurons (rat embryonic hippocampi, E18)	↓ Action potential ↑ Intracellular Ca^2+^ ↑ Synaptic activity (sEPSCs, sIPSCs)	[4]
	918 Hz 0.2 W/kg SAR	60 min	Primary astrocytes (rat neonatal brains and human fetal brains)	↓ ROS (mitochondrial) ↓ NADPH oxidase activity	[5]
	64/100 MHz, 0.4/0.6/0.9 W/kg SAR	1–2 h/day for 4, 8 or 14 days	Primary human brain cells (human fetal brain tissue)	↓ Aβ levels (Aβ40 and Aβ42, 64 MHz, 0.6 W/kg, 1 h/day for 14 days)	[6]
	1800 MHz 1/2/4 W/kg SAR	5 min on/10 min off for 1–3 days	Embryonic neural stem cells (mouse embryonic cortex, E13.5)	↓ Neurite number, branching points, and total length ↓ Proneural genes (*Ngn1*, *NeuroD* expression)	[7]
	50 Hz 1 MT	24 h	SH-SY5Y cells (human neuroblastoma cells)	↑ NOS activity ↑ O_2_^−^ production ↑ *TGF-β* and *IL-18BP* expression	[8]
	900 MHz 1 W/kg SAR	24, 48, 72, 120 h	SN56 cells (mouse cholinergic neurons) Primary cortical neurons (rat)	↑ β-thymosin mRNA ↓ Morphological maturation (neurites)	[9]
	1800 MHz 0.23 W/kg SAR	3 × 10 min/day for 2 days	SH-SY5Y cells (human neuroblastoma cells)	↑ ROS levels ↑ Monomeric α-syn levels ↑ Cell death	[11]
	1800 MHz 2 W/kg SAR	5 min on/10 min off for 24 h	Primary cortical neurons (newborn SD rats)	↑ ROS levels ↓ 8-OHdG levels ↓ Mitochondrial function ↓ *mtDNA* oxidative damage	[12]
	1900 MHz	2 h	Primary cortical neurons and astrocytes (ICR mouse embryonic, E15)	↑ Apoptotic pathways (*Caspase-2*, *Caspase-6*, *Asc*)	[13]
	2.45 GHz 6 W/kg SAR	20 min	N9 cells (mouse microglial cells)	↑ CD11b expression ↑ JAK2 and STAT3 phosphorylation ↑ Pro-inflammatory responses (*TNF-α* and *iNOS*)	[14]
	1950 MHz 6 W/kg SAR	2 h	HT22 cells (mouse hippocampal neuronal cells)	↑ ROS levels ↑ Cell death ↑ Neurotoxicity	[15]
	837 MHz (CDMA) 1950 MHz (W-CDMA) 2 W/kg SAR	2 h	HT22 cells (mouse hippocampal neuronal cells)	No effect on Aβ-induced cytotoxicity, ROS production, or apoptosis	[16]
	1950 MHz (W-CDMA) 6 W/kg SAR	2 h/day over 3 days	HT22 cells (mouse hippocampal neuronal cells) SH-SY5Y cells (human neuroblastoma cells)	No effect on the expression levels of *APP* and *BACE1* in either SH-SY5Y or HT22 cells.	[17]
	935 MHz 4 W/kg SAR	On/off cycles of 120/120 s	SH-SY5Y cells (human neuroblastoma cells) N9 cells (mouse microglial cells)	No effect on the proportions of living, early apoptotic, or late apoptotic cells in either SH-SY5Y or N9 cells.	[18]
	935 MHz 4 W/kg SAR	On/off cycles of 120/120 s for 24 h	SH-SY5Y cells (human neuroblastoma cells)	No effect on neuronal differentiation signaling pathways markers and mitochondrial fission and fusion markers	[19]
	1800 MHz 4 W/kg SAR	5 min on/10 min off for 1, 6, or 24 h	U251 and A172 cells (human glioblastoma) SH-SY5Y cells (human neuroblastoma cells)	No effect on cellular behavior. (cell cycle progression, cell proliferation, or cell viability)	[20]

“↑” (up arrow) indicates an increase, while “↓” (down arrow) indicates a decrease.

## 3. The Effects of Non-Contact RF-EMF Stimulation on Cognitive Behaviors and Molecular Mechanisms in In Vivo Animal Models

Several previous studies have reported potential beneficial effects on brain function as a result of RF-EMF exposure, particularly cognitive functions and behaviors in AD mouse models (Table 2) [21]. In *APP_SW_ + PS1* transgenic mice, RF-EMF exposure (918 MHz, 2 × 1 h/day for a month) increased mitochondrial function in brain regions such as the cerebral cortex and hippocampus. Improvement in mitochondrial respiration and ATP levels was not observed in the striatum or amygdala, but the enhanced function did affect regions involved in cognition (hippocampus and cortex) [22]. Various behavioral assessments demonstrated that RF-EMF exposure could mitigate cognitive impairments in AD mouse models. Notably, AD mice exposed to RF-EMF (918 MHz, 0.25 W/kg SAR, 2 × 1 h/day) in early adulthood were able to avoid some cognitive deficits, and older AD mice showed improvements in cognitive abilities in the radial arm water maze and Y-maze. Moreover, no harmful effects on sensorimotor function or anxiety were observed in these animals, as confirmed by open-field and elevated plus-maze tests [23]. In 5xFAD mice, long-term RF-EMF exposure (1950 MHz, 5 W/kg SAR, 2 h/day, 5 days/week for 6 months) decreased the deposition of Aβ and improved cognitive functions, although it did not change the expression levels of genes associated with Aβ processing [24]. Exposure to a higher-frequency EMF (1950 MHz, 5 W/kg SAR, 2 h/day, 5 days/week for 8 months) improved pathological behaviors such as anxiety and hyperactivity in a 5xFAD mouse model. Additionally, RF-EMF exposure increased glucose metabolism in the hippocampus and significantly reduced the size and number of Aβ plaques in the hippocampal CA1 region and entorhinal cortex, which are implicated in AD pathology [25,26]. RF-EMF exposure for periods longer than 8 months has been shown to improve cognitive abilities and behavior in multiple AD mouse models. These studies suggest that prolonged exposure could help counteract some of the behavioral and cognitive impairments associated with AD [22,25,27]. In another study, exposure to 2.4 GHz Wi-Fi type RF signals improved memory and reduced anxiety in 3xFAD mice, although there was no change in their motor activity in the flex field behavioral test [28].

Contradictorily to those beneficial and therapeutic results, some findings show that RF-EMF exposure has a lack of benefits or negative effects on brain function. In 5xFAD mice, RF-EMF exposure (1950 MHz, 5 W/kg SAR, 2 h/day for 5 days/week) did not produce any improvement in cognitive function or reduction in Aβ deposition, which are hallmarks of AD pathology, in brain sections from their hippocampi and cortices [29]. A study examining the effects of a specific type of electromagnetic pulse (EMP) exposure (100 Hz) found that it produced significant cognitive deficits in rats, as evidenced by their performance in the Morris water maze. Additionally, increased expression of amyloid precursor protein (APP) and Aβ oligomers in hippocampal neurons, as well as reduced antioxidant activity (SOD and GSH levels), were observed, suggesting that EMP exposure could contribute to cognitive decline and neurodegenerative changes [30,31]. Acute RF-EMF exposure also affected long-term contextual memory in rats [32]. Rats exposed to 900 MHz RF-EMF (15 min at 0, 1.5, and 6 W/kg SAR or 45 min at 6 W/kg SAR) exhibited reduced memory abilities compared with sham-exposed animals in behavioral assessments such as the Morris water maze. Notably, the expression of GFAP, an astrocyte marker, was increased in the hippocampus and olfactory bulb. These data suggest disruptions of glial cell functions that are critical for neural information processing, synaptic transmission, and neural activation [32]. Some studies also highlighted altered behavior and locomotor activity in animals exposed to RF-EMF. C57BL/6 mice exposed to 835 MHz RF-EMF (4 W/kg SAR for 5 h/day) for 12 weeks showed hyperactivity-like behavior in the rotarod test [33], and impaired calcium signaling in hippocampal neurons was also noted [34]. In *1-methyl-4-phenyl-1,2,3,6-tetrahydropyridine* (MPTP)-treated mice (a model of Parkinson’s disease), RF-EMF exposure impaired the recovery of locomotor activity after neural damage [35]. In rats exposed to RF-EMF (2.5 GHz) for 4, 6, and 8 weeks, a decrease in locomotor activity and increased anxiety levels were observed [36]. These data suggest that RF-EMF exposure might not have a therapeutic effect on neurodegenerative processes in AD, may not alleviate key pathological features, including Aβ accumulation in the brain, and may have negative impact on mental health-related behaviors. In C57BL/6 mice, RF-EMF exposure (1950 MHz, 5 W/kg SAR, 2 h/day, 5 days/week) for 8 months did not affect neuroinflammation or oxidative stress levels [37]. Exposure to 900 MHz RF-EMF (0.9 W/kg SAR, 2 h/day) for 45 days significantly increased ROS levels in rats [38], suggesting that RF-EMF exposure can induce oxidative stress, which is a well-known causative factor in neurodegenerative diseases such as AD. Rats exposed to 900 MHz RF (0.036 W/kg SAR, 3 h/day, 7 days/week) for 12 months showed decreased expression levels of *rno-miR-107* in brains. Since miRNAs as key regulators can have various biological impacts, and some miRNAs have been reported to be differentially expressed in AD. The decreased expression levels of *rno-miR-107* suggest that long-term RF-EMF exposure may increase the risk of neurodegenerative disease [39]. Daily exposure to high-intensity RF-EMF (2.5 GHz) for 6 and 8 weeks significantly reduced acetylcholinesterase (AChE) activity in the cerebral cortex, which can disrupt cholinergic neurotransmission [36]. Because AChE is critical to the acetylcholine neurotransmitter, which is involved in memory and learning, such changes could impair cognitive function. Similarly, exposure to 1800 MHz (4 h/day, 5 days/week) radiation for 90 days also produced a reduction in AChE activity [40], further supporting the idea that RF-EMF might negatively affect memory and learning processes.

Taken together, these studies indicate that the effects of RF-EMF exposure on brain functions are complex and potentially contradictory. Whereas some studies suggest no beneficial effects, others indicate that RF-EMF can impair memory, increase oxidative stress, disrupt neurotransmission, and alter behavior. Overall, the literature suggests that RF-EMF exposure, particularly at specific frequencies and durations, might have beneficial effects on cognitive function and behavior in AD models, potentially through mechanisms such as enhanced mitochondrial function, improved glucose metabolism, and a reduction in amyloid plaques. However, further research is needed to fully understand how the underlying mechanisms and effects of RF-EMF exposure on AD and related neurodegenerative conditions depend on factors such as the frequency, intensity, and duration of exposure.

## 4. The Effects of Other Non-Contact Electrotherapeutic Stimulation Modalities in In Vitro and Model Systems

In addition to RF-EMF stimulation, other electrical stimulation therapies, including visual stimulation, auditory stimulation, repetitive transcranial magnetic stimulation (rTMS), and transcranial electrical stimulation (tES), are being investigated as potential methods for modulating brain activity and improving cognitive function in vivo [21,41,42]. For example, visual stimulation such as light flicker stimulation, especially at 40 Hz, has been studied for its effects on brain oscillation (e.g., ɤ waves, 25–140 Hz) and its potential therapeutic role in neurodegenerative diseases such as AD. Similarly, auditory stimulation at specific frequencies (e.g., 40 Hz) has shown promise in reducing the amyloid burden and improving cognitive function. rTMS produces non-invasive magnetic pulses that modulate brain activity. Researches has suggested that it can improve attention, memory, and other cognitive functions. The tES technique uses electrical currents to influence brain excitability, with potential applications in treating cognitive impairments.

Frequencies in the range of 25–140 Hz, particularly around 40 Hz, have been linked to cognitive functions such as attention and memory. In AD models, ɤ oscillations are often disrupted, and restoring these rhythms through sensory stimulation (visual or auditory) has been shown to reduce markers of neuroinflammation and amyloid/tau accumulation, enhance mitochondrial function, and decrease the expression of apoptosis markers (*bcl-2*, *caspase-3*), potentially improving cognitive outcomes [43,44]. Exposure to 40 Hz visual stimulation (light flickering to stimulate the visual cortex) for 3 months reduced amyloid/tau levels and improved spatial learning (e.g., Morris water maze performance) in 3xFAD mice. Stimulation at 40 Hz (light or auditory) also reduced the amyloid burden in the brains of 5xFAD and *APP/PS1* mice [45,46,47]. In the GENUS study, ɤ entrainment with sensory stimuli (20 Hz, 40 Hz, or 80 Hz auditory or visual stimulation in 10 s blocks interleaved with 10 s baseline periods) for 7 days improved memory and reduced the amyloid burden in the hippocampus and auditory cortex [45,48]. In gamma electrical stimulation, 40 Hz RF-EMF exposure (1 h daily for 4 weeks) using a sinusoidal current resulted in microglial activation (as detected by *iba1*-positive cells) and improved learning and memory, with stronger effects at higher current intensities [48]. In contrast to those studies, a 40 Hz light flickering stimulus (12.5 ms on/off for 30 min) did not affect Aβ accumulation [49], suggesting variability in results depending on experimental conditions and model systems. Stimulation at 40 Hz and 100 Hz increased cell growth, neurite length, and differentiation in neuronal cell cultures, indicating that certain frequencies might promote cell proliferation and differentiation in SH-SY5Y cells [50].

Non-invasive electrical stimulation therapies, especially at specific frequencies such as 40 Hz, hold promise as potential treatments for cognitive dysfunction in neurodegenerative diseases such as AD. Although most studies show positive effects on amyloid reduction and cognitive improvement, some studies report mixed results, indicating that factors such as stimulation parameters (e.g., frequency, duration, intensity) and the specific animal models or in vitro systems used can significantly influence the outcomes (Table 3). Further research is needed to clarify the optimal conditions for these therapies and their clinical applicability.

## 5. The Effects of Non-Contact RF-EMF Stimulation on the Physiological Process of Sleep in Human Subjects

Studies have reported conflicting findings on the effects of RF exposure on human health. Whereas some studies suggest potential effects on brain activity, cognition, and sleep, others report little to no significant impact. An increasing number of studies are investigating the effects of EMFs on physiological functions. However, due to inconsistent results and the complexity of RF stimulation, the need for additional clinical trials is evident. Notably, a consistent finding is the alteration of electroencephalogram (EEG) patterns caused by RF exposure, particularly in the alpha and spindle frequency ranges. Remarkable experiments further reveal that RF exposure can influence brainwave activity during sleep, with a pronounced effect in the spindle frequency range (10.75–13.75 Hz). These changes are influenced by factors such as exposure intensity, duration, and whether the RF is targeted to a specific hemisphere of the brain. The effects were not uniform, as individual factors such as baseline brain activity, eye-opening status, and RF exposure intensity appeared to influence the extent of changes in EEG power.

Overall, several studies have reported an increase in power spectral density (PSD) following exposure to 900 MHz Global System for Mobile Communication (GSM) signals, which are characterized by harmonic pulses at 2 Hz, 8 Hz, 217 Hz, 1736 Hz, and higher frequencies [51,52,53,54,55,56]. In particular, an increasing number of studies have reported changes in the sleep spindle range, a distinct brainwave activity that occurs after the onset of non-REM sleep. Sleep spindles are characterized by an EEG frequency pattern of approximately 11–16 Hz. Although their exact function remains uncertain, research has established a strong correlation between sleep spindles and processes such as sensory integration and long-term memory consolidation [57]. Additionally, emerging evidence suggests that increased sleep spindle power contributes to improved sleep quantity and stability [58]. For instance, Schmidt et al. investigated sleep EEG changes by comparing the effects of 2 Hz pulse-modulated 900 MHz GSM RF exposure and magnetic field stimulation at a specific absorption rate (SAR) of 2 W/kg [59]. They found that RF stimulation increased PSD across the entire spindle frequency range, suggesting that brainwaves are influenced by a distinct mechanism unique to RF stimulation. In another experiment, Regel et al. exposed subjects to 900 MHz GSM signals at 0.2 and 5 W/kg SAR for 30 min prior to sleep [60]. The EEG data revealed increased power levels within the spindle frequency range, specifically at 10.75–11.25 Hz and 13.50–13.75 Hz. Similarly, Huber et al. observed that RF stimulation increased spindle frequency during sleep following exposure to 900 MHz GSM signals at 1 W/kg SAR for 30 min prior to sleep [54]. RF stimulation, randomly applied to either the left or right hemisphere, led to an increase in EEG activity across all frequency bands. Comparatively, the left hemisphere exhibited significantly greater increases in the 12.5–13.25 Hz and 9–13.5 Hz frequency ranges than the right hemisphere [61]. Additionally, EEG recordings showed an increase in power within the spindle frequency range (11.5–12.25 Hz) during non-REM sleep following 30 min exposure to 894.6 MHz GSM pulse-modulated radiation (0.11 W/kg SAR) prior to sleep [62]. Furthermore, Dalecki et al. investigated the impact of specific variables on the extent of spindle frequency increases induced by RF stimulation [63]. To explore this, they tested various conditions, including eye state (open or closed) and stimulation intensity, by applying 920 MHz GSM signals at 1 W/kg or 2 W/kg SAR to the left hemisphere. Consequently, the increase in alpha power in EEG data was more pronounced when the eyes were open, with stronger effects at higher stimulation intensities and during later periods of RF exposure. These findings suggest that a closed-eye state or insufficient exposure duration may explain the lack of detectable changes in alpha power following RF-EMF exposure.

Several studies have also demonstrated that RF stimulation primarily induces changes in EEG activity within the delta (0.5–4 Hz) and theta (4–8 Hz) frequency ranges, with minimal effects observed in the alpha (8–12 Hz) and beta (12–20 Hz) ranges. Lustenberger et al. investigated how individual variability impacts outcomes under controlled RF exposure conditions [64]. After RF-EMF exposure for 30 min before sleep (900 MHz pulsed at 2 Hz, 2 W/kg SAR), they found power increases in the delta–theta frequency range at frontocentral electrodes, indicating localized changes in brain activity. However, these changes did not lead to significant alterations in overall sleep-related EEG parameters, suggesting that the effects were localized to specific frequency ranges rather than influencing overall sleep. This finding underscores the importance of considering inter-individual variability, exposure parameters, and specific physiological metrics when evaluating the effects of RF-EMF on sleep and brain activity. It also highlights the potential for non-uniform responses, emphasizing the complex interplay between external stimulation and intrinsic individual factors.

Sleep is not merely a state of rest, but a fundamental biological process essential for cognitive functions such as learning and the consolidation of long-term memory. During sleep, particularly in stages like non-REM sleep, the brain actively processes and integrates information acquired during wakefulness, solidifying neural connections critical for memory retention. Recognizing this role of sleep, researchers aimed to explore how RF exposure might influence these cognitive processes. To this end, they measured participants’ ability to learn new information and retain memories following sleep, seeking to understand whether RF-induced alterations in sleep patterns or brain activity might disrupt or enhance these vital cognitive functions. First, Lustenberger et al. demonstrated that RF exposure enhances slow-wave activity (0.75–4.5 Hz) during sleep, while higher-frequency bands, such as 12–15 Hz, remain unaffected. This increase in slow-wave activity was associated with a negative impact on motor task performance. In their experiment, which involved 16 participants, a motor sequence task was used to evaluate memory retention. The task required repetitive learning of a finger-tapping sequence, providing a measure of how RF exposure influenced cognitive and motor learning processes [65]. Conversely, Bueno-Lopez et al. reported no significant changes in EEG activity following non-pulsed 2450 MHz RF exposure [66]. To further investigate the effects of RF exposure on sleep, the study incorporated sleep-related memory experiments alongside EEG measurements. Declarative memory was evaluated by having participants learn 101 word pairs before sleep, followed by an assessment of their ability to quickly and accurately recall the pairs upon waking (*n* = 30). When RF exposure (6.4 mW/kg SAR) was applied during an 8 h sleep period, the researchers observed significant improvements in declarative memory [66]. These findings suggest that RF stimulation may influence brainwave activity and human behavior, with effects varying based on stimulation methods, participant conditions, frequency bands, and environmental factors [67]. Notably, changes in EEG power were observed in various frequency ranges, particularly within the delta–theta and spindle frequency bands. However, not all observed changes were associated with significant alterations in sleep architecture or cognitive performance. When Lowden et al. investigated self-evaluated sleepiness and objective electroencephalogram architecture after RF exposure to 1930–1990 MHz, 3G UMTS signals (1.6 W/kg SAR, 3 h/day) for 2 days, sigma (11–12.75 Hz) power activity was reduced during sleep periods. As is known, the sigma band (11–15 Hz) is associated with sleep spindle activity. However, no significant changes were found in delta, theta, or other frequency bands [68]. These results emphasize the need for further research to elucidate the mechanisms by which RF stimulation affects sleep and brain function.

The variability in these findings points to the need for more controlled, long-term studies. Factors such as RF exposure parameters (frequency, power, duration) and individual characteristics (e.g., baseline EEG activity) must be carefully considered. Understanding of the specific mechanisms through which RF exposure influences brain activity remains a major gap in the current literature. Further research is also needed to identify whether these effects are transient, cumulative, or potentially harmful.

## 6. The Effect of RF-EMF Stimulation on the Pathophysiology of the Human Brain

In addition to its effects on sleep, numerous studies have examined the impact of RF exposure on other physiological and cognitive processes, including nociception, cognitive performance, cerebral glucose metabolism, and auditory function (Table 4). First, Vecsei et al. investigated the effects of RF exposure on nociception, focusing on how exposure influences the perception and processing of pain signals within the nervous system [69]. Participants were exposed to a 2140 MHz UMTS signal at a specific absorption rate (1.62 W/kg SAR) for 30 min, applied to their right ear. Following the exposure, a finger heat pain threshold test was conducted. The results indicated no change in the baseline heat pain threshold. However, under the sham condition, a desensitization effect was observed, characterized by an increased pain threshold after the second pain stimulus, which typically makes the stimulus feel less painful. In contrast, this desensitization effect was absent under the RF exposure condition. Interestingly, only under actual RF exposure did participants report a slight decrease in subjective pain perception, suggesting that RF exposure may alter pain cognition or processing.

Emerging evidence suggests that RF stimulation can influence cognitive functions that involve complex brain activities. Regel et al. investigated the effects of RF stimulation on healthy participants, exposing them to either pulse-modulated or continuous-wave GSM RF (900 MHz, 1 W/kg SAR) for 30 min [70]. Participants who received pulse-modulated GSM stimulation demonstrated a decrease in response speed during working memory tasks, accompanied by an improvement in task accuracy. Moreover, an increase in EEG power within the 10.5–11 Hz range was observed 30 min after exposure. These effects were absent in participants exposed to continuous-wave RF stimulation. Interestingly, no effects on cognitive abilities were observed when participants were stimulated with 2.14 GHz RF, in contrast to the effects observed with 900 MHz stimulation [71]. The authors compared the effects of continuous-wave and pulse-modulated UMTS waves at a frequency of 2.14 GHz (1.5–2.2 V/m). After exposure to each condition, participants underwent cognitive testing, and no significant differences in cognitive performance were observed between the two stimulation types. These findings suggest that the impact of RF stimulation on cognitive function may depend on specific characteristics of the stimulation, particularly whether it is pulse-modulated and the frequency range utilized.

In addition, several studies indicate that RF stimulation applied to the user’s head ultimately influences brain glucose metabolism by modulating neuronal excitability [72,73,74]. In 2006, Ferreri et al. examined the impact of GSM mobile phone emissions on brain excitability [74]. In a double-blind crossover study with 15 male participants, exposure to a 900 MHz GSM signal for 45 min significantly altered intracortical excitability. Specifically, short intracortical inhibition was reduced and intracortical facilitation was enhanced in the exposed hemisphere compared to the non-exposed hemisphere or sham exposure. These findings suggest that RF-EMFs can modulate neuronal excitability, potentially influencing neural function. Thereafter, Volkow et al. explored the effects of acute mobile phone RF-EMF exposure on brain glucose metabolism, a marker of neural activity, in a randomized crossover study with 47 participants [73]. Using positron emission tomography (PET) scans, the study revealed no significant changes in whole-brain metabolism. However, regions closest to the phone’s antenna, such as the orbitofrontal cortex and temporal pole, exhibited increased glucose metabolism following 50 min of exposure to 900 MHz RF-EMF. This localized effect highlights the need for further investigation into the clinical relevance of RF-induced metabolic changes. Recently, Wardzinski et al. investigated the relationship between RF-EMF exposure and human food intake behavior in a randomized crossover study with 15 healthy males [75]. After 25 min of exposure to RF-EMFs from mobile phones, participants demonstrated a significant increase in total caloric intake, particularly from carbohydrates, compared to sham exposure. Furthermore, 31-phosphorus magnetic resonance spectroscopy (31P-MRS) revealed increased cerebral energy markers, such as ATP levels, suggesting that RF-EMF may disrupt brain energy homeostasis and promote overeating. These findings propose a potential link between RF exposure and metabolic regulation, with implications for understanding the obesity epidemic.

Moreover, Maby et al. investigated the effects of RF exposure on auditory processing by exposing participants to a 900 MHz GSM signal (1.4 W/kg SAR) alongside auditory stimuli and measured auditory evoked potentials (AEPs) [76]. AEPs, which are recordings of neural responses to auditory stimuli captured as brainwave signals, were analyzed to assess changes induced by RF stimulation. Their findings revealed alterations in the correlation coefficients of the AEPs during RF exposure. However, further analysis of auditory cortical activity through scalp localization suggested that while RF exposure influenced auditory evoked potentials, there was no conclusive evidence linking these changes to functional impairments in brain activity [77].

**Table 4 ijms-26-02268-t004:** The effects of RF-EMF at a clinical level.

Non-Contact Stimulation Exposure					
Stimulation Type	Frequency and Intensity	Exposure Periods	Participant	Effects	References
RF-EMF	900 MHz 1 W/kg SAR	15 min on/15 min off during 8 h sleep episode	Healthy young males (mean age: 22.6 years)	↑ Non-REM sleep EEG power	[51]
	900 MHz 1 W/kg SAR	30 min before a 3 h sleep episode	Healthy young males (mean age: 20–25 years)	Short exposure to EMF emitted by mobile phones affects brain physiology.	[52]
	900 MHz 1 W/kg SAR	30 min	Healthy young males (mean age: 20–25 years)	↑ Regional cerebral blood flow in dorsolateral prefrontal cortex (rCBF) ↑ Alpha activity (EEG during wakefulness)	[53]
	900 MHz 1 W/kg SAR	15 min on/15 min off during 8 h sleep episode/ 30 min before a 3 h sleep episode	Healthy young males	↑ Non-REM sleep EEG power	[54]
	894.6 MHz 0.11 W/kg SAR	30 min before sleep episode	Healthy individuals (males and females)	↑ Rapid eye movement sleep	[55]
	900 MHz	15–20 min	Healthy young males and children	Induce short-term, reversible changes in human EEG	[56]
	900 MHz 2 W/kg SAR	30 min before sleep episode	Healthy young males (mean age: 23.2 years)	Affects non-rapid eye movement sleep and rapid eye movement sleep activity	[59]
	900 MHz 0.2/5 W/kg SAR	30 min before sleep episode	Healthy young males (mean age: 22.4 years)	Affects the non-REM sleep EEG and cognitive performance	[60]
	900 MHz 1 W/kg SAR	25 min	Healthy individuals (males and females; mean age: 18–30 years)	Exposure may affect human brain activity	[61]
	894.6 MHz 0.11 W/kg SAR	30 min before sleep episode	Healthy individuals (males and females; mean age: 27.9 years)	Affects the subsequent EEG spectral power during non-REM sleep	[62]
	920 MHz 0/1/2 W/kg SAR	30 min	Healthy individuals (males and females; mean age: 24.4 years)	The alpha power increases when the eyes open than eyes close EEG during RF-EMF exposure	[63]
	900 MHz 2 W/kg SAR	30 min before sleep episode	Healthy young males (mean age: 23.3 years)	No effect on sleep-related EEG activity	[64]
	900 MHz 0.15 W/kg SAR	Over the night	Healthy young males (mean age: 19.9 years)	Affects brain activity during sleep and may interfering with cortical excitability renormalization and synaptic plasticity	[65]
	2.45 GHz 6.4 mW/kg SAR	8 h sleep episode	Healthy young males (mean age: 24.12 years)	No effect on EEG changes, but may improve declarative memory	[66]
	3.5 GHz 0.037 ± 0.011 mW/kg (HASAR) 0.008 ± 0.019 mW/kg (BASAR)	2 h	Healthy individuals (males and females; mean age: 26.6 years)	No effect on the EEG activity in healthy adults	[67]
	1930–1990 MHz (3G UMTS) 1.6 W/kg SAR	3 h/day) for 2 days	Healthy individuals (males and females; mean age: 18.6 years)	↓ Sigma (11–12.75 Hz) power activity	[68]
	900 MHz 1.4 W/kg SAR	-	Healthy individuals (males and females; mean age: 25 years)	Affects the correlation coefficients of the auditory evoked potentials (AEPs)	[76]
	2140 MHz (UMTS)	45 min	Healthy individuals (males and females; mean age: 37.7 years)	No effect on well-being or cognitive performance	[77]
	900 MHz 1 W/kg SAR	30/60 min	Healthy young males (mean age: 22.1 years)	Transiently affects cognitive performance and brain activity	[70]
	2.14 GHz (CW, UMTS)	45 min	Healthy individuals (adolescents and adults; mean age: 15–16 years; 25–40 years)	No effect on cognitive performance in healthy adolescents or adults	[71]
	900 MHz 0.97/1.33 W/kg SAR	25 min	Healthy young males (mean age: 23.47 years)	Enhances the ATP synthesis rates, thereby increasing carbohydrate intake.	[75]

“↑” (up arrow) indicates an increase, while “↓” (down arrow) indicates a decrease.

## 7. Discussion

In this review, we discussed the effects of RF exposure on in vitro, in vivo, and human brain activities. The studies about the potential effects of RF exposure on brain function are very limited and the experimental results are controversial. Specifically, the variability in individual responses, as well as the diverse exposure conditions (e.g., intensity, duration, and frequency of RF signals), contribute significantly to the complexity of drawing definitive conclusions.

Numerous studies suggested that RF-EMF has potential neuroprotective effects in Aβ-induced neurodegenerative environments like AD [1,2,21]. RF exposure has been shown to reduce reactive oxygen species (ROS) production [5] and to enhance neuronal excitability and synaptic transmission [3,4]. RF-EMF reduced the number of Aβ plaques and improved cognitive function in the AD animal models [5,6,22,25,26,27]. Several human studies suggested that RF exposure might potentially influence cognitive function, with individual differences [60].

On the other hand, some studies have reported negative or no effects of RF exposure on the nervous system. RF-EMF increased oxidative stress levels [11], and promoted cell apoptosis, exacerbating the pathology of neurodegenerative diseases [13,14,30,31]. Some studies found that RF exposure did not significantly change Aβ deposition or cognitive function [18,19,29].

Therefore, we suggest that further studies with controlled methodologies will shed more light on these variations and help establish a clearer understanding of RF exposure’s impact on brain function to be used for therapeutic purposes in pathological conditions, including neurological or psychiatric diseases. In addition, more short- or long-term clinical trials are needed to verify the effectiveness and safety of RF exposure in cognitive functions to provide more scientific basis and practical guidance for future therapeutics.

## 8. Conclusions

In conclusion, RF exposure has the potential to affect neural stimulation and influence various brain activities in in vitro and in vivo models. The in vitro/in vivo effects of RF-EMF exposure are summarized in Figure 1. RF-EMF exposure therapy might improve cognitive performance in optimized conditions. Cognitive dysfunction caused by increased reactive oxygen species and oxidative stress may be improved after RF-EMF exposure through cellular mechanisms such as mitochondrial restoration, gene expression regulation, and cytoskeletal trafficking, etc. Recent studies have explored the influence of EMF on human physiology, particularly brain activity, cognition, sleep, behavior, and sensory functions. Research data vary depending on the RF stimulation conditions, highlighting the need for more clinical trials to clarify its potential effects. However, research is still very limited, and conflicting results can arise depending on the exposure conditions and individual variations. Moreover, although extensive research has been conducted to assess the effects of RF exposure, current data remain insufficient to understand its biological impact. Therefore, careful consideration is needed before clinically applying RF exposure. Additionally, further investigation into the effects of RF and their mechanisms is essential, as many researchers strive to establish reliable safety and efficacy data.

## Figures and Tables

**Figure 1 ijms-26-02268-f001:**
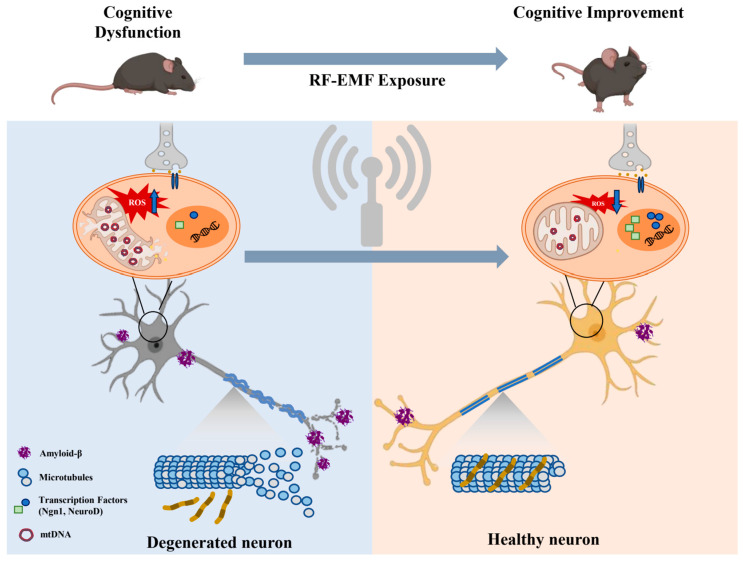
Summary of the in vitro/in vivo effects of RF-EMF exposure. With optimized exposure parameters and timing, RF-EMF exposure therapy might improve cognitive performance. Cellular mechanisms of neuronal responses to RF-EMF exposure are schematically summarized. Cognitive dysfunction caused by increased reactive oxygen species and oxidative stress may be improved after RF-EMF exposure through mitochondrial restoration, gene expression regulation, and cytoskeletal trafficking. “↑” (up arrow) indicates an increase, while “↓” (down arrow) indicates a decrease.

**Table 2 ijms-26-02268-t002:** The effects of RF-EMF in in vivo animal models.

Non-Contact Stimulation Exposure					
Stimulation Type	Frequency and Intensity	Exposure Periods	Animal Models (Gender, Age)	Effects	References
RF-EMF	918 MHz 0.25/1.05 W/kg SAR	2 × 1 h/day for 1 month	*APPsw/PS1* (Tg) (15–17 months)	↑ Cognitive behavior (radial arm water maze) ↑ ATP production (147–159%) ↓ ROS level ↑ Complex IV activity (1133% in cortex, 1158% in hippocampus)	[22]
	918 MHz 0.25 W/kg SAR	2 × 1 h/day for 2–8 months	*AβPP/PS1* (Tg) (2–13 months)	↑ Cognitive behavior (radial arm water maze and Y-maze) ↓ Aβ plaque burden (hippocampus and cortex)	[23]
	1950 MHz 5 W/kg SAR	2 h/day, 5 days/week for 6 months	5xFAD (Tg) (female, 6–8 months)	↑ Cognitive behavior (novel object recognition test and Y-maze) ↓ Aβ deposition No effect on the expression levels of genes associated with Aβ processing	[24]
	1950 MHz 5 W/kg SAR	2 h/day, 5 days/week for 8 months	5xFAD (Tg) (female, 1.5 months)	↑ Cognitive behavior (novel object recognition test) ↓ Anxiety-like behavior (open field test) ↑ Glucose metabolism (hippocampus and amygdala)	[25]
	1950 MHz 5 W/kg SAR	2 h/day, 5 days/week for 8 months	5xFAD (Tg) (female, 1.5 months)	↓ Aβ40 and Aβ42 levels (hippocampus and cortex) ↓ APP and BACE1 expression ↓ GFAP and Iba1 expression ↑ Memory performance (passive avoidance test and Y-maze)	[26].
	918 MHz 0.25/1.05 W/kg SAR	2 h/day for 2 months	*APPsw/PS1* (Tg) (21–27 months)	↑ Cognitive behavior (Y-maze) ↓ Aβ plaque burden (hippocampus and cortex) ↑ Aβ disaggregation ↑ Energy metabolism ↓ Oxidative stress	[27]
	2400 MHz 1.6 W/kg SAR	2 h/day for 4 weeks	3xTg-AD (male, 12 months)	↑ Cognitive behavior (Barnes maze) ↓ Anxiety-like behavior (two-compartment box test)	[28]
	1950 MHz 5 W/kg SAR	2 h/day, 5 days/week for 3 months	5xFAD (female, 1.5 months)	No effect on behavioral performance (Y-maze, Morris water maze, novel object recognition test, and open field test)	[29]
	100/1000/10,000 pulses Do not show SAR value	Only one single exposure	Sprague Dawley rats (male, 2 months)	↓ Cognitive behavior (Morris water maze) ↓ Aβ expression ↓ SOD activity and GSH content ↑ LC3-II expression	[30]
	100/1000/10,000/100,000 pulses	1–1000 s/day for 8 months	Sprague Dawley rats (male, 2 months)	↓ Cognitive behavior (Morris water maze and Y-maze) ↑ Anxiety-like behavior (open field test and elevated plus maze) ↑ Aβ levels ↑ Oxidative stress (SOD, GSH, MDA) ↑ LC3-II expression	[31]
	900 MHz 0/1.5/6 W/kg SAR	0, 1.5, or 6.0 W/kg for 15 min or 6.0 W/kg for 45 min	Sprague Dawley rats (male, 6 weeks)	↑ GFAP levels (striatum, 1.5 W/kg SAR) ↑ Cytosolic GFAP levels (hippocampus and olfactory bulb, 6 W/kg SAR) ↓ Cognitive behavior (fear conditioning test, 6 W/kg SAR)	[32]
	835 MHz 4 W/kg SAR	5 h/day, 5 days/week for 12 weeks	C57BL/6 mice (male, 6 weeks)	↑ Locomotor activity (open field test) ↑ Beclin1, LC3B-II expression ↓ Bax and Bcl2 protein level	[33]
	835 MHz 4 W/kg SAR	5 h/day for 4 weeks	C57BL/6 mice (male, 6 weeks)	↓ Voltage-gated calcium channel expression ↓ Bax ↑ Autophagy-related genes levels (*Atg5, Atg9A, Beclin2, LC3B*)	[34]
	835 MHz 4 W/kg SAR	5 h/day for 12 weeks	C57BL/6 mice (male, 6 weeks)	↓ Synaptic Vesicle (SV) density ↓ Dopamine Levels ↓ TH Expression ↓ Locomotor activity (open field test and rotarod test) ↓ Synapsin I/II levels	[35]
	2.5 GHz Do not show SAR value	24 h/day for 4, 6 and 8 weeks	Albino rats (male, 4 weeks)	↓ Exploratory behavior (open field test) ↓ Locomotor activity (rotarod test) ↓ AChE enzymatic activity ↑ AChE mRNA expression	[36]
	1950 MHz 5 W/kg SAR	2 h/day, 5 days/week for 8 months	C57BL/6J (female, 14 months)	No effect on oxidative stress, DNA damage, neuroinflammation, or apoptosis	[37]
	900 MHz 0.9 W/kg SAR	2 h/day for 45 days	Wistar rats (male, 35 days)	↓ Antioxidant enzyme activity (GPx, SOD) ↑ Catalase activity ↑ ROS levels ↓ Protein kinase C (PKC) ↓ Melatonin Levels ↓ Apoptosis (caspase-3) ↑ Creatine kinase (CK)	[38]
	900 MHz 0.036 W/kg SAR	3 h/day, 7 days/week for 12 months	Wistar rats (male)	↓ *rno-miR-107* expression	[39]
Microwave	1800 MHz 0.433 W/kg SAR	4 h/day, 5 days/week for 90 days	Wistar rats (male)	↑ Oxidative stress (GSH) ↑ IL-6 and TNF-α expression ↑ DNA damage ↓ AChE activity	[40]

“↑” (up arrow) indicates an increase, while “↓” (down arrow) indicates a decrease.

**Table 3 ijms-26-02268-t003:** The effects of other non-contact electrotherapeutic stimulation in in vitro and in vivo model systems.

Non-Contact Stimulation Exposure					
Stimulation Type	Frequency and Intensity	Exposure Periods	Animal Models/ Cell Line	Effects	References
Light Flickering	40 Hz	6 days/week for 12 weeks	3xTg-AD (male, 15 months)	↓ Aβ and tau levels ↑ Cognitive behavior (Morris water maze, step through avoidance test) ↑ Bax and cleaved caspase 3 expression ↓ Bcl-2 expression	[44]
	40/80 Hz	1 h/day for 7 days	3xTg-AD (female, 6 months)	↓ Aβ40 and Aβ42 levels ↑ Synaptophysin (PSD-95)	[47]
	40 Hz	1 h (Acute exposure) 1 h/day for 7 days (Chronic exposure)	*APP/PS1* (male and female, 5–12 months) 5xFAD (male and female, 4–7 months)	No effect on AD pathology in *APP/PS1* and 5xFAD mice.	[49]
Multi-Sensory Gamma Stimulation	40 Hz (Auditory and visual stimulation)	1 h/day for 7 days (visual flicker stimulation) 20 min (Auditory tone train stimulation)	5xFAD (male, 6 months)	↓ Amyloid plaques (neocortex) ↑ Cognitive behavior (Morris water maze, novel object recognition)	[45]
	8/40/80 Hz (Auditory and visual stimulation)	1 h	5xFAD (male and female, 6 months)	↓ Amyloid plaques (40 Hz)	[46]
Gamma electrical stimulation	40 Hz (25/50/100/200 µA)	1 h/day for 4 weeks	5xFAD (male, 3 months)	↓ Aβ40 and Aβ42 levels ↑ Microglia cell counts ↑ Cognitive behavior (Morris water maze)	[48]
Low-magnitude low-frequency (LMLF) vibrations	40/100 Hz	8 h/day for 5 days	SH-SY5Y cells (human neuroblastoma cells)	↑ Length of neurites ↑ Differentiation Levels	[50]

“↑” (up arrow) indicates an increase, while “↓” (down arrow) indicates a decrease.

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
