# Peer review of "Brain Disease-Modifying Effects of Radiofrequency as a Non-Contact Neuronal Stimulation Technology"

_ijms, 2025, doi:10.3390/ijms26052268_

Round 1

Reviewer 1 Report

Comments and Suggestions for Authors

To the Authors

The Ms entitled “Brain Disease-Modifying Effects of Radiofrequency as a Non-Contact Neuronal Stimulation Technology” (Ms ID ijms-3429392) focuses on the effects of RF exposure on brain function. This is a growing area of research, explored, for obvious reasons, either in negative or in positive. Overall research results are highly dependent on the RF stimulation conditions, thus highlighting the need of adequate clinical trials to clarify its real potential effects.

Major point

 The current review is quite accurate and adds perspective to the knowledge in the field. Nonetheless, I do agree with the Authors that, at this time, research on RF exposure on brain function is quite limited and inconclusive, with conflicting results highly dependent on the exposure conditions and individual variations. Although the Ms organization and writing accuracy are appreciable, the above mentioned research limitations, of course, do also coincide with the very limits of the present review article itself.

Minor points

1.    In order to slightly enhance the Ms. impact, I would suggest to include a paragraph with a more comprehensive discussion of the body of knowledge gathered so far by the Authors, and to shorten up the Conclusion section to 1-2 paragraphs.  

2.    I would also suggest the Authors to devote some more summarizing paragraphs delineating possible perspectives and future lines of research in the RF exposure studies.  

3.    Refs list is apparently accurate and up-to-date although approximately 1 in 5 citations is really recent, i.e. published in the last 5 years despite at least 3 cited papers having been published in the last couple of years (years 2023 to 2024). Hence, the Authors are encouraged to further update the Refs List.   

Reviewer 2 Report

Comments and Suggestions for Authors

This is an interesting and carefully organized review of the literature on radio-frequency stimulation of the brain in animals and humans. I am not an expert in this field, but I found the systematic treatment of these many and varied studies clearly presented.

I have some suggestions the authors might consider to help the reader in interpreting the meaning of the review. 

It appears to me there are two very different typies of studies: (1) studies attempting to improve brain function (or remove toxic metabolites) with RF treatments, and (2) studies attempting to assess the potentially deleterious effects of RF stimulation subsequent to cell phone usage. Is this correct?  If so, these might be separated and labeled as such, given that there is an obvious bias that differs between the studies (looking for therapeutic effects versus looking for damage). It is indeed useful to examine the effects of RF stimulation regardless of the intention or expected risk/benefit, but rather than just listing all these studies and their effects, a clearer rationale for reviewing them together would help the reader make sense of the findings.

Could the authors help the readers compare the actual stimulation paramters between the therapeutic and the cell-phone-toxicity studies? A strength is the specification of the parameters of RF frequency and power, but a non-RF-engineer will not be able to make these comparisons easily. My impression is that the frequencies and parameters between the two classes of studies is so different that comparisons are not really possible.  Is that correct? 

Reporting the quantitative changes in the EEG is the obvious first step, but providing further interpretive comments on the functional health significance may be warranted in some cases. For spindle changes, this is difficult: spindles are necessary for sleep health and memory consolidation, yet non-organized spindle activity may be reflective of disorganization or impairment. Can the authors draw on the literature on sleep spindles a little more carfully to address this question?

The increase of delta and theta activity following RF stimulation is more concerning as an indication that the stimulation may be causing neural disruption or damage.  EEG slowing (greater delta and theta) is a sign of brain dysfunction in many conditions, such as hepatic encephalopathy, closed head injury, and dementia. This is not a definitive marker, and it's not impossible that greater delta and theta could reflect sleepiness or drowsiness induced by the stimulation. Although providing such interpretation is clearly challenging, to the extent that the authors can draw from existing clinical literature to raise these questions of the meaning of the observed effects this paper would have more impact in interpreting the meaning of this important and extensive literature.

Round 2

Reviewer 1 Report

Comments and Suggestions for Authors

To the Authors

The Authors of Ms ID ijms-3429392 have fully and satisfactorily addressed my prior issues of concern.

The Ms is improved in terms of scientific communication of its content.
